# Generalizability, Robustness and Replicability When Evaluating Wellbeing of Laboratory Mice with Various Methods

**DOI:** 10.3390/ani12212927

**Published:** 2022-10-25

**Authors:** Dietmar Zechner, Benjamin Schulz, Guanglin Tang, Ahmed Abdelrahman, Simone Kumstel, Nico Seume, Rupert Palme, Brigitte Vollmar

**Affiliations:** 1Rudolf-Zenker-Institute of Experimental Surgery, Rostock University Medical Center, 18057 Rostock, Germany; 2Unit of Physiology, Pathophysiology and Experimental Endocrinology, Department of Biomedical Sciences, University of Veterinary Medicine Vienna, 1210 Vienna, Austria

**Keywords:** distress of animals, suffering, severity assessment, 3Rs, animal welfare science

## Abstract

**Simple Summary:**

It is in the interest of the general public as well as the scientific community to optimize the wellbeing of animals during scientific research. To reach this goal, methods need to be defined which can reliably evaluate the wellbeing of animals. In this study, we assessed whether various methods, such as measuring body weight, burrowing activity, nesting behavior, a distress score and fecal corticosterone metabolites can differentiate between healthy mice and mice after surgical intervention or during the progression of a gastrointestinal disease. The ability of each method to differentiate between these two states of wellbeing was different between distinct surgical interventions and gastrointestinal diseases. These data suggest that scientists cannot rely on a single method, but have to combine many methods when assessing the wellbeing of animals.

**Abstract:**

An essential basis for objectively improving the status of animals during in vivo research is the ability to measure the wellbeing of animals in a reliable and scientific manner. Several non-invasive methods such as assessing body weight, burrowing activity, nesting behavior, a distress score and fecal corticosterone metabolites were evaluated in healthy mice and after three surgical interventions or during the progression of four gastrointestinal diseases. The performance of each method in differentiating between healthy and diseased animals was assessed using receiver operating characteristic curves. The ability to differentiate between these two states differed between distinct surgical interventions and distinct gastrointestinal diseases. Thus, the generalizability of these methods for assessing animal wellbeing was low. However, the robustness of these methods when assessing wellbeing in one gastrointestinal disease was high since the same methods were often capable of differentiating between healthy and diseased animals independent of applied drugs. Moreover, the replicability when assessing two distinct cohorts with an identical surgical intervention was also high. These data suggest that scientists can reach valid conclusions about animal wellbeing when using these methods within one specific animal model. This might be important when optimizing methodological aspects for improving animal wellbeing. The lack of generalizability, however, suggests that comparing animal models by using single methods might lead to incorrect conclusions. Thus, these data support the concept of using a combination of several methods when assessing animal welfare.

## 1. Introduction

Since Russell and Burch proposed the 3Rs (replacement, reduction and refinement of animal experiments) as principles for humane experimental procedures in 1959 [1], scientists and governments have adopted and expanded these principles. The passing of Directive 2010/63/EU in 2010 made the assessment of animal wellbeing in scientific procedures mandatory in the European Union [2]. In a similar fashion, animal welfare regulations are implemented and enforced by Animal Care and Use Committees in the United States [3]. While the legal framework in all member states of the EU explicitly demands prospective and retrospective assessment and classification of the severity of procedures, scientists struggle to implement objective, evidence-based and validated methods to assess animal wellbeing in the face of ever-expanding quantities of animal models for diseases. 

In general, the assessment of animal wellbeing is often based on physiological parameters such as body weight [4,5,6] or fecal corticosterone metabolites (FCMs) [7,8,9,10,11]. In addition, clinical signs of distress [12,13]) or behavior such as nesting [14,15,16,17,18] burrowing [18,19,20,21] or wheel-running [22,23] are also often assessed as indicators for wellbeing of rodents. Although it is believed that a combination of different methods yields a more precise conclusion than relying on single parameters [24,25,26], these single parameters should ideally be sensitive enough on their own to discriminate between distressed and non-distressed animals. Thus, it is of interest to check each method if it reliably detects distress when study design (different disease models or therapeutic interventions) or input data (different datasets, baseline data) are varied. In clinical settings, receiver operating characteristic (ROC) curves are often used to measure the performance of a diagnostic test [27,28,29,30]. These curves are generated by plotting sensitivity (true positive rate) versus 1-specificity (false positive rate) using a range of thresholds. A diagonal line indicates no discriminatory power (diseased versus non-diseased) of the diagnostic test. This diagonal line equals an area under the curve (AUC) of 0.5. An AUC of 1 indicates that a diagnostic test has 100% sensitivity and 100% specificity and, therefore, has perfect discriminatory power. The AUC was originally also described as corresponding to the probability of classifying a randomly chosen diseased subject as diseased with a higher suspicion than a random non-diseased subject [31]. Thus, the AUC is a measure of the discriminative ability of prediction models [32]. The AUC can serve as an easy-to-use metric to define and compare the diagnostic ability of different methods to differentiate between two states (e.g., healthy versus diseased). As scientists and governmental agencies would like to base their decisions on robust methods, which can define animal wellbeing with a high replicability, it is an important step in the area of animal welfare science to compare these methods for their diagnostic capabilities.

In accordance with Goodman et al. [33] and the Subcommittee on Replicability in Science [34], we define the term replicability as follows: “Replicability refers to the ability of a researcher to duplicate the results of a prior study if the same procedures are followed but new data are collected. That is, a failure to replicate a scientific finding is commonly thought to occur when one study documents relations between two or more variables and a subsequent attempt to implement the same operations fails to yield the same relations with the new data” [34]. For example, replicability is given when a second experiment applies the same procedures and confirms the conclusion of a first experiment. Robustness refers to the stability of experimental conclusions to variations in either baseline assumptions or experimental procedures [33]. For example, a method for assessing animal wellbeing would be robust when the same conclusion can be reached and when minor methodological changes are implemented. Robustness is related to the concept of generalizability, which refers to the persistence of an effect in settings different from and outside of an experimental framework [33]. For example, a method for assessment of animal wellbeing would have very high generalizability when it can measure animal wellbeing in diverse animal models.

In this study, we applied ROC curves to assess if burrowing activity, nesting activity, changes in body weight, a distress score and FCMs can distinguish between healthy and distressed animals after diverse surgical interventions or during the progression of four different gastrointestinal diseases. In extension of this goal, we evaluated the replicability of conclusions when using distinct cohorts of animals. Furthermore, we assessed how robust conclusions are when using different baseline data, different methods of presenting data or using different therapeutic interventions (vehicle or therapy). This compilation of data should also give a first impression on the generalizability of these methods, when assessing various surgical interventions and gastrointestinal diseases. Please note that this project was not started with a clearly defined hypothesis. Thus, the following data interpretation is exploratory rather than confirmatory research. 

## 2. Materials and Methods

### 2.1. Animals

#### 2.1.1. Study Concept and Animal Husbandry

This study did not use new animals, but re-evaluated data generated for previous projects with the novel focus on summarizing and comparing the performance of distinct methods when differentiating between healthy mice and mice after surgical intervention or during a disease. Mice, which had to be euthanized during the experiment, were excluded from the analysis. During the experiments, all mice were kept single-housed in type III cages (Zoonlab GmbH, Castrop-Rauxel, Germany) at a 12 h light–dark cycle (dark: 7 pm–7 am), a temperature of 21 ± 2 °C and relative humidity of 60 ± 20% with food (pellets, 10 mm, ssniff-Spezialdiäten GmbH, Soest, Germany) and tap water available ad libitum. Enrichment was provided by nesting material (shredded tissue paper, Verbandmittel GmbH, Frankenberg, Germany), a paper roll (75 × 38 mm, H 0528–151, ssniff-Spezialdiäten GmbH) and a wooden stick (40 × 16 × 10 mm, Abedd, Vienna, Austria). The health of the animal stock was routinely checked (Helicobacter sp., *Rodentibacter pneumotropicus*, and murine Norovirus were detected in few mice; these animals were not used for any experiments). All animal experiments were approved by the German local authority: Landesamt für Landwirtschaft, Lebensmittelsicherheit und Fischerei Mecklenburg-Vorpommern (-1-062/16, -1-019/15, and -1-002/17).

#### 2.1.2. Surgical Interventions and Induction of Diseases

For the transmitter implantation, methodological details and some data were published previously [35]. In brief, male C57BL/6J mice (age: 17.3/17.0–17.45, median/interquartile range in weeks) were anaesthetized on day 0 with isoflurane, a midline laparotomy was performed and an ETA-F-10 transmitter (Data Sciences International, St. Paul, MN, USA; weight: 1.6–1.7 g) was placed in the abdominal cavity. The negative electrode was lead subcutaneously to the right pectoralis major muscle, where it was fixed by sutures. The positive electrode was guided subcutaneously to the left side and was sutured onto the external oblique muscle. The peritoneum and the skin lesion was closed with sutures as described previously [35]. The surgical procedure took 45–50 min.

For the pancreatic cancer model, male C57BL/6J mice (age: 18.6/18.6–19.7, median/interquartile range in weeks) were anaesthetized on day 0 with isoflurane, the abdominal cavity was opened by laparotomy and 5 μL of a cell suspension containing 2.5 × 10^5^ 6606PDA cells was injected slowly into the pancreas using a 25-μL syringe (Hamilton Syringe, Reno, NV, USA). The abdominal cavity was closed with sutures as described previously [36]. The surgery lasted 15–20 min. Starting on day 4, and after cell injection, mice were intraperitoneally injected on a daily basis with either metformin (Met; 125 mg/kg in phosphate buffered saline) and α-cyano-4-hydroxycinnamate (CHC; 15 mg/kg in 50% dimethylsulfoxide) or the corresponding vehicle solutions until euthanasia on day 37. More methodological details and some data were published previously [36].

For the ligation of the bile duct, methodological details and some data were published previously [26]. In brief, male BALB/cANCrl mice (age: 10.9/9.6–13.7, median/interquartile range in weeks) were anaesthetized on day 0 with isoflurane and the abdominal cavity was opened by laparotomy. The common bile duct was ligated by three surgical knots and was then transected between the two distal ligations. The abdominal cavity was closed by absorbable sutures and the skin lesions were sewed using a prolene suture. The surgical procedure took 25–40 min. In order to evaluate the possible therapeutic efficacy of NLRP3, inflammasome inhibitor MCC950 (Sigma Aldrich, St. Louise, MO, USA), 20 mg/kg MCC950 or aqua (vehicle) was intraperitoneally injected (volume: 10 µL/g body weight) daily from day 1 before BDL to day 13 after BDL. The mice were euthanized on day 14 after bile duct ligation.

When inducing intoxification with carbon tetrachloride, male BALB/cANCrl mice (age: 10.0/7.9–10.4 median/interquartile range in weeks) were intraperitoneally injected on days 0, 4, 7, 11, 14, 18, 21, 25, 28, 32, 35 and 39 with 0.25 mL/kg body weight CCl_4_ (Merck Millipore, Eschborn, Germany, code 1.02209.1000, volume: 1 µL/g body weight after 4x dilution with corn oil). The experiment ended on day 42. Methodological details and some data were published previously [26].

Chronic pancreatitis was induced with cerulein (Bachem, H-3220.0005, Bubendorf, Switzerland), which was dissolved in 0.9% sodium chloride and was administered by consecutive intraperitoneal injections (dosage: 50 μg/kg, volume: 5 µL/g body weight, three hourly injections/day; three days/week (on days 0, 2, 4, 7, 9, 11, 14, 16, 18, 21, 23, 25, 28 and 30) into male C57Bl/6J mice (age: 15.3/14.7–15.3 median/interquartile range in weeks). The microRNA-21 inhibitor (miRCURY LNA™ microRNA-21a-5p inhibitor; cat. # 339203 YCO0070656, sequence: TCAGTCTGATAAGCT) and its corresponding control (miRCURY LNA™ microRNA-21a-5p control; cat. # 339203 YCO0070657, sequence: TCAGTATTAGCAGCT) were purchased from Qiagen (Hilden, Germany), resuspended in PBS and injected subcutaneously at a dosage of 10 mg/kg (volume: 5 µL/g body weight) on day 0 and day 14. The experiment ended on day 33 after the first cerulein injection. Methodological details and some data were published previously [37].

The following refinement measures were implemented. Before surgical intervention, a single subcutaneous injection of 5 mg/kg carprofen (Rimadyl^®^, Pfizer, GmbH, Berlin, Germany) was applied (volume: 2µL/g body weight) and the eyes of the mouse were kept wet by using eye ointment (Jenapharm, Jena, Germany). During as well as after surgical intervention, the mice were warmed by a warming plate or a warming lamp. In all experiments, 1250 mg/L metamizol (Ratiopharm, Ulm, Germany) was provided daily in the drinking water until euthanasia was performed on the animals. 

### 2.2. Assessment of Animal Wellbeing

In order to evaluate animal wellbeing, the body weight, burrowing activity, nesting behavior, the distress score and FCMs were assessed for each mouse at distinct time points. All these parameters were evaluated at two time points before (pre 1, pre 2) and directly after surgical intervention (post). For example, the distress score was evaluated on day 0 (30 min after finishing surgery), burrowing and nesting activity was assessed from the evening of day 0 to the morning of day 1 and on day 1 after surgery, body weight was determined and feces were collected (see Appendix A). In order to get an overview of the wellbeing of animals during the progression of a disease, all parameters were assessed during the early (cholestasis: day 1–2; CCl_4_ intoxication: day 4–5; pancreatic cancer: day 4–8; chronic pancreatitis: day 2–3), middle (cholestasis: day 4–5; CCl_4_ intoxication: day 18–19; pancreatic cancer: day 18–19; chronic pancreatitis: day 16–17) and late phase (cholestasis: day 13–14; CCl_4_ intoxication: day 39–40; pancreatic cancer: day 34–35; chronic pancreatitis: day 30–31) of each disease.

The burrowing activity was analyzed using a tube (length: 15 cm, diameter: 6.5 cm) filled with 200 g of food pellets [18,19]. The tube was placed into the mouse cage 2–3 h before the dark phase and the remaining pellets were weighed after 2 h (for C57Bl/6J mice) or 17 ± 2 h (for BALB/cANCrl mice). 

To analyze nest-building behavior, a cotton nestlet (5 cm square of pressed cotton batting, Zoonlab GmbH, Castrop-Rauxel, Germany) was placed into the cage 30 to 60 min before the dark phase. The nests were scored in the morning of the following day at 9:30 ± 2 h, by using a scoring system developed by Deacon [18]. However, a 6th score point was added to this scoring system. This score defined a perfect nest: The nest looked like a crater and more than 90% of the circumference of the nest wall was higher than the body height of the coiled-up mouse.

In addition, the wellbeing of mice was evaluated by assessing multiple parameters with the help of a distress score sheet. This score sheet was based on other score sheets [5,38] and previously published by our group [39]. The score summarizes various defined criteria (e.g., spontaneous behavior, flight behavior, or general body conditions).

In order to assess the concentration of fecal corticosterone metabolites [7], feces dropped within 24 h in the home cage were collected, dried for 4 h at 65 °C and stored at −20 °C. Afterwards, 50 mg of dry feces were extracted with 1 mL 80% methanol for subsequent analysis using a 5α-pregnane-3β,11β,21-triol-20-one enzyme immunoassay [7,10,40,41]

### 2.3. Data Presentation and Statistical Analysis

Graphs and all biostatistical analysis were done using GraphPad Prism8 (GraphPad Software Inc., San Diego, CA, USA). To determine how well a parameter distinguishes between healthy and diseased animals, we used the ROC curve analysis and determined the area under the curve (AUC) with corresponding 95% confidence intervals (CI) as a measurement for the performance of the methods. In addition, this software gives the asymptotic *p*-value that determines if the AUC is significantly different from an AUC of 0.5 (an AUC of 0.5 suggests no discriminative ability of a diagnostic test). For examples of ROC curve analysis and explanatory notes, see Appendix A. GraphPad Prism computes a *p*-value (two tailed) using the z ratio, which was calculated using the equation z = (A − 0.5)/SE. Since the *p*-value considers both the AUC and data variability, we used the *p*- value for giving a representative overview in the form of heat maps. Differences with *p* = 0.01–0.05 were considered to be significant, and differences with *p* < 0.01 were considered to be highly significant.

## 3. Results

We first assessed animal wellbeing before and after a common surgical intervention using the intraperitoneal implantation of a telemetric transmitter. When comparing the body weight on the day after transmitter implantation (post) to a day before transmitter implantation (pre 1), a reduction in body weight was observed (Figure 1A). The ROC curves, measuring the performance of this method in differentiating between these two states of animal wellbeing (healthy animals versus animals after surgical intervention), yielded an AUC of 0.90 with a 95% confidence interval of 0.75–1.00 (Figure 1A). The discriminatory power of this method was, therefore, significantly higher (*p* = 0.0028) than methods without any discriminative power, yielding an AUC of 0.5. When choosing another day as baseline (pre 2), the same conclusion with a similar AUC was reached (Figure 1B). 

Another way of presenting data is comparing the percentage in body weight change between the two time points, (pre 1−pre 2) × 100/pre 2, before surgical intervention, and the percentage in body weight change between the day after surgical intervention and the first time point, (post−pre 2) × 100/pre 2. Transmitter implantation caused a reduction in body weight (Figure 1C), and the performance of this method reached a very high AUC of 1.00 with a 95% confidence interval of 1.00–1.00 (Figure 1C). The discriminatory power was, therefore, also significantly higher (*p* = 0.0002) than methods without any discriminative power. All three ways of calculating changes in body weight led to the same conclusion that measuring body weight could differentiate quite well between animals before and after implantation of a transmitter. However, the third method had the highest discriminative power.

In a similar manner, we evaluated burrowing activity, nesting behavior, a distress score and FCMs for their discriminative power to differentiate between animals before and after transmitter implantation. Burrowing activity, nesting behavior, the distress score and FCMs could very well discriminate between animals before versus after telemeter implantation (see Appendix A). 

The same evaluations were done with two other surgical interventions, laparotomy followed by injection of cancer cells into the pancreas or laparotomy followed by bile duct ligation. In order to give an overview on the discriminative power of body weight, burrowing activity, nesting activity, the distress score and FCMs on all three surgical interventions, we plotted the *p*-values in a heat map (Figure 2) and presented AUC, 95% confidence interval and the number of data points analyzed in Appendix A. All methods (body weight change, burrowing, nesting distress score and FCMs) had a very high discriminative power (*p* < 0.01), when differentiating between animals before and after transmitter implantation (Figure 2). However only two methods, the evaluation of burrowing activity and assessing FCMs, had a very high discriminative power (*p* < 0.01) when differentiating between animals before and after cell injection into the pancreas (Figure 2). Three methods (the evaluation of burrowing activity, nesting behavior and assessing a distress score) had a very high discriminative power (*p* < 0.01) that differentiated between animals before and after bile duct ligation (Figure 2). A fourth method, assessing the body weight of the animals, had only a very high discriminative power, when comparing the percentage in body weight change. Comparing the body weight of mice as raw data (measured in grams) before and after bile duct ligation had, however, very low discriminative power (*p* > 0.05). 

These results indicate that some of these methods lack generalizability, because their suitability to differentiate between healthy and distressed animals was dependent on the specific surgical intervention. Alternatively, it might also be possible that the predictive power of the methods cannot be replicated, even when the same surgical intervention is performed.

To assess this hypothesis, we analyzed two groups of mice before and one day after a laparotomy with cell injection into the pancreas. Since treatment with a drug started four days after cell injection, these groups were treated as identical during data collection. For this surgical intervention, a high (*p* = 0.01–0.05) or very high (*p* < 0.01) discriminative power was observed in both groups, group A and group B, for the evaluation of burrowing activity and FCMs (Figure 3A; for AUC and confidence interval see Appendix A). Thus, the conclusion, based on which methods can differentiate between healthy animals and animals after this surgical intervention, can be replicated. However, a high variability in the *p* value was sometimes observed in these experiments, possibly because we only analyzed a few (*n* = 7) mice. As a next step, we assessed if methods can differentiate between healthy and distressed animals in a robust manner when an identical surgical intervention has been performed but different therapies were applied. Both cohorts had the identical surgical intervention, a bile duct ligation, but were intraperitoneally injected either with MCC950 or a vehicle solution (Figure 3B; for AUC and confidence interval see Appendix A). A high (*p* = 0.01–0.05) or very high (*p* < 0.01) discriminative power was observed for the evaluation of body weight change, burrowing activity, nesting behavior and assessment of the distress score when differentiating between animals before and after bile duct ligation, independent of which cohort was analyzed (Figure 3B). 

We then assessed animal wellbeing before and during various gastrointestinal diseases. For each gastrointestinal disease, we compared two different cohorts, one treated with a specific drug and the other cohort with the respective vehicle solution. Change in body weight, burrowing activity, nesting behavior and the distress score had high (*p* = 0.01–0.05) or very high (*p* < 0.01) discriminative power when discriminating between healthy animals and animals during bile duct ligation-induced cholestasis (Figure 4; for AUC and confidence interval see Appendix A). This was observed in the animal cohort treated with MCC950 and in the animal cohort treated with the respective vehicle solution. However, only one method, evaluation of nesting behavior, had very high (*p* < 0.01) discriminative power when discriminating between healthy animals and animals with CCl_4_-induced liver fibrosis (Figure 4). Again, this was observed in the animal cohort treated with MCC950 and in the cohort treated with the vehicle solution. None of the tested methods could differentiate well between healthy animals and animals with pancreatic cancer (Figure 4). This was observed in both cohorts of animals, independent of the animals that were treated with α-cyano-4-hydroxycinnamate and metformin or the vehicle solution. When analyzing an animal model for chronic pancreatitis, the change in body weight, burrowing activity and nesting behavior had high (*p* = 0.01–0.05) or very high (*p* < 0.01) discriminative power when differentiating between healthy animals and mice suffering from chronic pancreatitis (Figure 4). FCMs had only significant discriminative power when mice with chronic pancreatitis were treated with the vehicle solution, but not when they were treated with the microRNA-21 inhibitor (Figure 4).

We then assessed animal wellbeing separately during three different time points, the early, middle and late phases of each gastrointestinal disease. Change in body weight, burrowing activity, nesting behavior and the distress score had high (*p* = 0.01–0.05) or very high (*p* < 0.01) discriminative power when discriminating between healthy animals and cholestatic animals at all phases of cholestasis (Figure 5). However, only nesting behavior had high (*p* = 0.01–0.05) or very high (*p* < 0.01) discriminative power when differentiating between healthy animals and animals with CCl_4_-induced liver fibrosis (Figure 5; for AUC and confidence interval see Appendix A). In the animal model for pancreatic cancer, body weight had high (*p* = 0.01–0.05) discriminative power only during the middle and late phase of cancer progression, whereas burrowing activity had very high (*p* < 0.01) discriminative power only during the early phase of cancer progression (Figure 5). In the animal model for chronic pancreatitis, changes in body weight, burrowing activity and nesting behavior had very high (*p* < 0.01) discriminative power when differentiating between healthy and diseased animals. However, FCMs only had high (*p* = 0.01–0.05) or very high (*p* < 0.01) discriminative power in the early and late phases of chronic pancreatitis.

## 4. Discussion

The presented data suggest that the replicability of conclusions about the wellbeing of animals when analyzing body weight, burrowing, nesting, the distress score or FCMs can be high. Moreover, the robustness of these methods when varying calculation methods or drug treatment can also be quite high. However, the generalizability of these methods, when used for different animal models, seems to be low.

Several limitations in the interpretation of these data exist. First of all, ROC curves do have advantages but also some limitations. An advantage is that ROC curves can analyze ordinal, non-continuous data [42]. This was essential for the analysis of non-continuous data, such as the distress score or the score for nesting activity. However, ROC analysis with few and unevenly distributed ordinal data may cause unreliable estimations [42]. We observed few and unevenly distributed data points especially when evaluating the distress score. Thus, future work should critically evaluate, how reliable ROC analysis is when describing the diagnostic ability of the distress score. In addition, we want to emphasize that we analyzed three to five methods simultaneously to differentiate between healthy and diseased animals without correcting for the accumulation of the alpha error. This might also contribute to an overestimation of the benefit of these methods when differentiating between healthy and diseased animals. Another limitation is that ROC curves derived from small-sample data sets may not always reliably reflect a classifier’s true performance [43]. The sample size for reliably determining if the method is better than random guessing depends on the AUC and the allocation ratio [44]. For example, for an AUC = 0.95 (a Type I error = 0.05, a power of 0.8, and an allocation ratio of 3) *n* = 3 and *n* = 9 data points, whereas for an AUC = 0.65 (a Type I error = 0.05, a power of 0.8, and an allocation ration of 3) *n* = 29 and *n* = 87 data points are suggested [44,45]. Thus, methods described by ROC curves with low AUC will especially benefit from a higher number of data points.

Another major limitation of this study, but also of many studies in the area of animal welfare, is that one measures differences between healthy animals and animals after a surgical intervention or induction of a disease. Any difference observed is often interpreted as proof of reduced wellbeing. One should be aware that this is only one possible interpretation. This limitation is especially evident when only one method supports the interpretation of reduced wellbeing. For example, when repetitively injecting CCl_4_ into mice, only nesting behavior was reduced (Figure 5). No reduction of body weight, burrowing activity and no increase in the distress score was noticed. This suggests that nesting activity is either the most sensitive method to detect distress or CCl_4_ causes a specific change in nesting behavior independent of inducing distress. Consistent with the second hypothesis is the observation that CCl_4_ can change protein expression, reduce the number of neurons in the brain [46] and can also lead to brain damage [47]. Such changes in the central nervous system could influence complex behavior such as nesting activity.

Another limitation is that only one data set described replicability (Figure 3A). It demonstrates that two out of five methods (burrowing and FCMs) can differentiate between the state of the animals before and after an surgical intervention and that this conclusion can be replicated with a second set of data (Figure 3A). One could attempt to quantify replicability by describing that the conclusion concerning all five methods was replicated to 100%. Thus, between the two sets of experiments, the identical methods had or did not have discriminative ability between healthy mice and mice after this surgical intervention. However, please note that this conclusion is based on *p* < 0.05, which determines if the AUC is significantly different from an AUC of 0.5. Such a *p*-value is a completely arbitrary threshold [48] and there is the danger of over-interpreting conclusions based on such an arbitrary *p*-value [49,50,51]. However, this approach helps to simplify observations and to present a simplified overview in the form of a heat map in order to notice patterns in complex data and suggest hypotheses, which can be verified or falsified in future experiments. This experiment suggests that an intra-laboratory replicability can be given for a certain experimental set up (Figure 3A). It does not demonstrate that replicability is given for all other surgical interventions. 

Several data sets support the concept that a high robustness is given, when different ways of defining the baseline during an experiment are evaluated (Figure 1, Figure 2 and Appendix A). Using data measured on different days before a surgical intervention did not have a major influence on the conclusion, or whether a method can differentiate between the distress level before and after a surgical intervention. However, when analyzing a laparotomy with bile duct ligation, there was a difference when using raw data or the percentage in body weight change between two time points (Δ% compared to pre 1 or pre 2 in Figure 2). Using the percentage in body weight change was very well suited to differentiate between distress levels before and after surgical intervention, whereas using raw data (body weight in g) did not differentiate well between these two levels of distress. Please note that in this experiment, animals with a high variability in age were used (age: 10.9/9.6–13.7, median/interquartile range in weeks). Thus, we suggest that the percentage in body weight change rather than raw data should be used, especially when evaluating the distress of animals of different ages and therefore different body weights. 

Several data sets support the conclusion that these methods can differentiate well between two levels of distress in a robust manner when treating animals, for example, with a drug or control vehicle (Figure 3B and Figure 4). Only one exception was noticed when analyzing FCMs during chronic pancreatitis (Figure 4). Analyzing FCMs could differentiate with a high (*p* = 0.0477) discriminative power between healthy mice and mice with chronic pancreatitis when these mice were treated with a vehicle solution. However, FCMs failed to differentiate between these two states of distress (*p* = 0.1992) when the mice were treated with an miR-21 inhibitor. As mentioned above, such a conclusion is based on *p* < 0.05, an arbitrary threshold. Because *p* = 0.0477 is very close to this arbitrary threshold, one should avoid over-interpreting this result. This overall robustness to small changes in the experimental protocol suggests that these methods are well suited to assess measures of refinement during animal experiments.

When evaluating, if a method can discriminate between healthy and diseased animals at different disease phases, the robustness for some methods was high, but for other methods it was lower (Figure 5). For example, nesting activity had a high or low discriminative power in all four animal models independent of the phase analyzed (100% robustness). Burrowing, however, was or was not a good method to discriminate in all phases only in three of four animal models (75% robustness). These data demonstrate that the capability of methods to measure differences between healthy and diseased animals can vary during the course of a disease. 

The generalizability of some methods was surprisingly low. This was observed when analyzing three different surgical interventions, but also when assessing four gastrointestinal diseases (Figure 2 and Figure 4). For example, nesting behavior could differentiate well between healthy mice and mice after a laparotomy with transmitter implantation and after a laparotomy with bile duct ligation (Figure 2). However, nesting behavior could not differentiate well between healthy mice and mice after a laparotomy with cancer cell injection. The same was observed when using the distress score (Figure 2). To explain these results, one could assume that a laparotomy with cancer cell injection causes less distress than the other two surgical interventions and that nesting behavior and the distress score are not sensitive enough to measure these low levels of distress. Sometimes, the length of a surgical intervention can be a good indication for the complexity of the intervention and for the complications after the intervention [52]. Indeed, a laparotomy with cancer cell injection was done within 15–20 min, whereas a laparotomy and transmitter implantation or a laparotomy with bile duct ligation took 45–50 min or 25 to 40 min, respectively. This supports the hypothesis that a laparotomy with cancer cell injection might cause less distress than the other two surgical interventions. However, the concept that nesting behavior has very low sensitivity when detecting the distress of animals seems to be in contrast to data presented in Figure 4. Thus, it supports the hypothesis described above, whereby CCl_4_ might reduce nesting activity directly without reducing animal welfare.

When analyzing gastrointestinal diseases, nesting behavior had the highest generalizability of all methods tested, since it could actually discriminate between healthy and diseased animals in three out of four animal models (75%), whereas burrowing activity could only discriminate in two out of four animal models (50%). 

A low generalizability of methods could also be explained by the assumption that certain methods only work well in specific mouse strains. However, strain specificity of certain methods cannot explain the observed lack of generalizability in this study. For example, analyzing change in body weight, burrowing activity and the distress score discriminated well between healthy and diseased animals when analyzing cholestasis in BALB/c mice. These methods failed, however, during CCl_4_ intoxication using the same mouse strain. Similarly, nesting could discriminate well between healthy mice and mice suffering from chronic pancreatitis, but failed in mice with pancreatic cancer. Both experiments were done with C57Bl/6 mice. 

The presented data demonstrate that different animal models can have diverse effects on various read out parameters for distress. We suggest that mainly two aspects influence whether these methods can differentiate between healthy and diseased mice in an animal model. One major aspect is the level of distress experienced by an animal during the experiment. A second aspect might be that some experiments might influence certain read out parameters and that this is independent from the distress experienced by an animal. Both aspects might reveal a biologically valid mechanism. This emphasizes the need to evaluate distress with several methods and not rely on one or few methods for describing animal wellbeing. However, we do not know yet what the methods with the highest generalizability to assess distress are and how many methods suffice for a correct evaluation of animal wellbeing. A few concepts have been tested to combine multiple methods to reach a conclusion about animal welfare. For example, one can perform such an analysis using z-scores [53], k-means clustering [54], principal component analysis [55,56] binary logistic regression [36,37] or support vector machine classification [26]. Another recently developed tool for a multivariate analysis of animal wellbeing is the Relative Severity Score (RELSA), which was developed by Talbot et al. [25], and is currently tested by various research groups [57]. In this study, we gave an overview on the ability of several methods to differentiate between healthy and diseased animals. We hope that the evaluation of additional animal models and methods will clarify within the following years, which methods are most suitable to assess and compare animal welfare. When distress is assessed, one key issue will be the question as to when the distress was actually measured. Please note that it was our intention to always measure the maximal level of distress. For example, we assessed the distress score always 30 min after an intervention, at a time point when a high impact on the distress score was observed (see Appendix A). The body weight, however, was measured 24 h after the intervention because a reduction in body weight can best be observed in the morning of the following day. We do not dare to predict yet if such first steps towards a data-based evaluation of animal welfare will ever allow us to define maximally allowed thresholds of distress an animal should be allowed to experience in a scientific, non-arbitrary manner.

## 5. Conclusions

We conclude that ROC curves can evaluate the performance of certain methods when differentiating between healthy and diseased mice. We suggest that this approach should also be conducted when evaluating other methods (e.g., running activity or mouse grimace scale). ROC curves or similar strategies might also be useful for assessing welfare of other species, such as pigs or sheep. However, for each species, different methods might be especially valuable for assessing the welfare of animals. Assessing the replicability, robustness and generalizability of the performance of certain methods in many different laboratories could provide a basis for deciding which methods are most suitable for multivariate analyses of animal wellbeing. Defining methods, which are highly replicable and robust might support scientists when assessing measures of refinement or when comparing the severity of animal models. However, so far, we have only analyzed a limited number of animal models. There is still the need to explore the robustness of methods in additional to animal models. Moreover, we also know little about the influence of sex and genetic background of animals on distinct methods or inter-personal as well as inter-laboratory differences when assessing animal wellbeing. 

## Figures and Tables

**Figure 1 animals-12-02927-f001:**
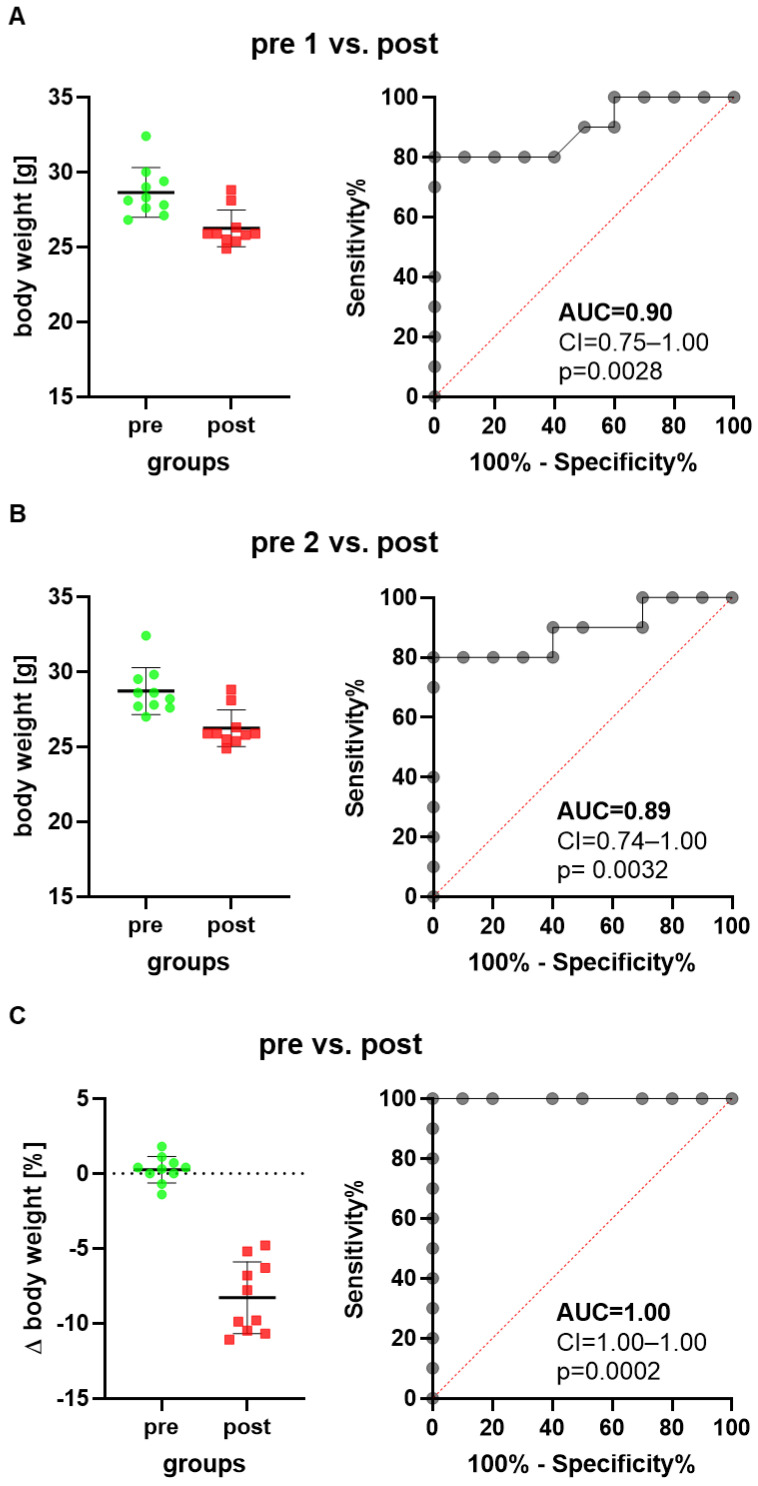
Scatter plots and ROC curves describe changes in body weight of mice when implanting a telemetric transmitter. The ROC curves are generated by plotting sensitivity (true positive rate) versus 1-specificity (false positive rate). The body weight one day after transmitter implantation (post) is compared to the body weight 5 days before transmitter implantation, presented as timepoint pre 1 (**A**), or to the body weight 2 days before transmitter implantation, presented as time point pre 2 (**B**). The percentage in body weight change between the two days before transmitter implantation is compared to the percentage in body weight change between the postoperative day and pre 1 (**C**). The classifier performance of this method in differentiating between animals before and after implanting a transmitter was characterized by the area under the curve (AUC), the confidence interval (CI) and the *p* value indicating how significant the difference was to the reference line (red dotted line indicating no discriminative power). *n* = 10 mice.

**Figure 2 animals-12-02927-f002:**
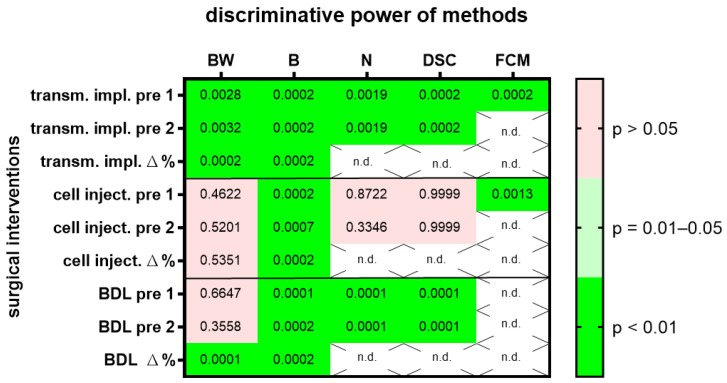
The heat map presents the *p*-value, which indicates how significant the discriminative power of each method was when differentiating between animals before and after surgical intervention. A laparotomy with transmitter implantation (transm. impl.), cancer cell injection into the pancreas (cell inject.) or bile duct ligation (BDL) was performed. The change in body weight (BW), burrowing activity (B), nesting behavior (N), a distress score (DSC) and fecal corticosterone metabolites (FCMs) was assessed. Animal wellbeing after transmitter implantation was compared to a timepoint, pre 1, or to another time point, pre 2, before implantation with most methods. In addition, the percentage of body weight change and burrowing activity between the two days before transmitter implantation is compared to the percentage in body weight change between the postoperative day and pre 1. The heat map differentiates between no (*p* > 0.05), a high (*p* = 0.01–0.05) and a very high (*p* < 0.01) discriminative power. Data analysis was not done (n.d.) for percent calculations of non-metric data (N, DSC) or for FCMs after BDL, since corticosterone metabolites need a functional bile duct so that it can be assayed in the feces. Transmitter implantation: *n* = 10 mice. Cell injection: *n* = 14. BDL: *n* = 14 for N (*n* = 7 vehicle-treated and *n* = 7 MCC950-treated) and *n* = 16 for BW, B and DSC (*n* = 9 vehicle-treated and *n*= 7 MCC950-treated).

**Figure 3 animals-12-02927-f003:**
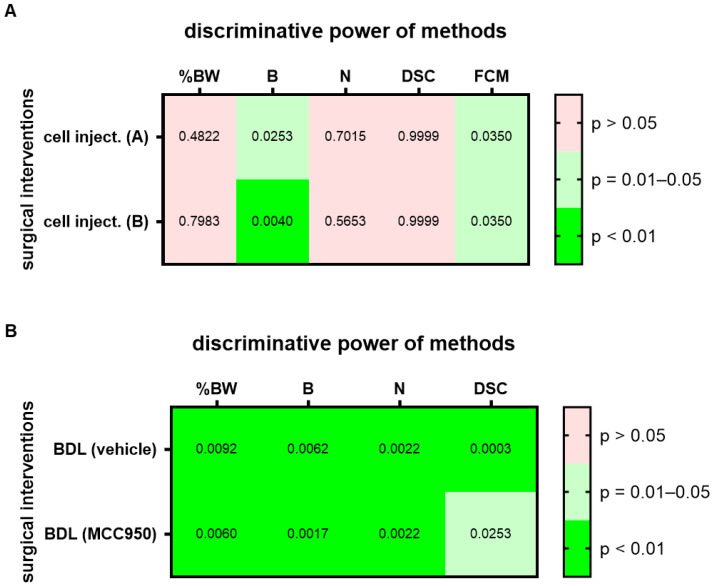
Replicability and robustness of the discriminative power of each method when assessing animal welfare. The heat map indicates the discriminative power when differentiating between animals before and after cell injection (cell inject.) into the pancreas (**A**). Both groups of animals (group A and B) received identical treatment. The discriminative power when differentiating between animals before and after bile duct ligation (BDL) is also presented in form of a heat map (**B**). Mice were treated with either a drug (MCC950) or vehicle solution (vehicle). The percentage in body weight change (%BW), burrowing activity (B), nesting behavior (N), a distress score (DSC) and fecal corticosterone metabolites (FCMs) was assessed. The heat maps present the *p*-values differentiating between no (*p* > 0.05), high (*p* = 0.01–0.05) and very high (*p* < 0.01) discriminative power between animals before and after surgical intervention. Since corticosterone is metabolized in the liver, no FCMs were analyzed after BDL (n.d.). For cell injection of cohort A (treated at a later time point with vehicle): *n* = 7. For cell injection of cohort B (treated at a later time point with CHC and Met): *n* = 7. For BDL and vehicle treatment: *n* = 7 mice (for N) or *n* = 9 (for %BW, B and DSC). For BDL and MCC950 treatment: *n* = 7 (for %BW, B, N and DSC).

**Figure 4 animals-12-02927-f004:**
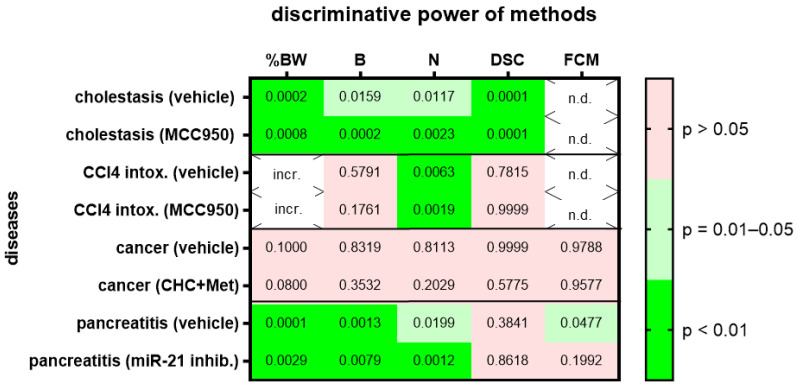
Heat map presenting the *p*-value, which indicates how significant the discriminative power of each method was when differentiating between animals before and after induction of a disease (pooling the data from early, middle and late disease phase). Two groups of mice with cholestasis, intoxification by CCl_4_, pancreatic cancer or chronic pancreatitis were either treated by the indicated drugs or by a solution lacking the drug (vehicle). The percentage in body weight change (%BW), burrowing activity (B), nesting behavior (N), the distress score (DSC) and fecal corticosterone metabolites (FCMs) was assessed. The heat map differentiates between no (*p* > 0.05), high (*p* = 0.01–0.05) and very high (*p* < 0.01) discriminative power. Since corticosterone is metabolized in the liver, no FCMs were analyzed during diseases with liver damage (n.d.). Since mice showed an increase rather than a decrease in body weight after treatment with CCl_4_, we did not consider this to be an indication of reduced animal welfare (incr.) and did not show the data. Mice used for cholestasis: *n* = 7 (vehicle-treated for N), *n* = 9 (vehicle-treated for %BW, B and DSC), *n* = 7 (MCC950-treated for %BW, B, N and DSC). Mice used for CCl4 intoxification: *n* = 6 (vehicle-treated for N), *n* = 3 (vehicle-treated for %BW, B and DSC), *n* = 6 (MCC950-treated for N) and *n* = 7 (MCC950-treated for %BW, B and DSC). Mice used for pancreatic cancer: *n* = 7 (vehicle-treated for %BW, B, N, DSC and FCM), *n* = 7 (CHC and Met treated for %BW, B, N, DSC and FCM). Mice used for chronic pancreatitis: *n* = 8 (vehicle-treated for %BW, B, N and DSC), *n* = 8 (miRNA21 inhibitor treated for %BW, B, N, DSC and FCM).

**Figure 5 animals-12-02927-f005:**
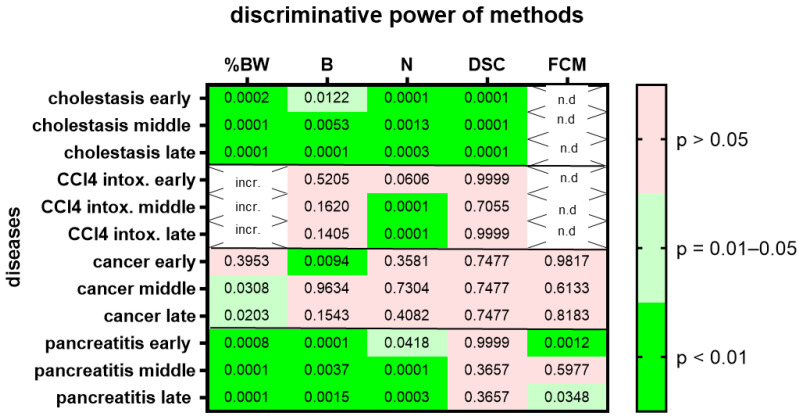
Heat map presenting the *p*-value, which indicates how significant the discriminative power of each method was when differentiating between animals before and at the early middle or late phase of a disease (pooling the data from mice treated with drugs and mice treated with vehicle). The percentage in body weight change (%BW), burrowing activity (B), nesting behavior (N), distress score (DSC) and fecal corticosterone metabolites (FCMs) was assessed. The heat map differentiates between no (*p* > 0.05), high (*p* = 0.01–0.05) and very high (*p* < 0.01) discriminative power. Since corticosterone is metabolized in the liver, no FCMs were analyzed during diseases with liver damage (n.d.). Since mice showed an increase rather than a decrease in % body weight after treatment with CCl_4_, we did not consider this to be an indication of reduced animal welfare (incr.) and decided not to present the *p*-value. Mice used for cholestasis: *n* = 14 for N (*n* = 7 vehicle-treated and *n* = 7 MCC950-treated), *n* = 16 for %BW, B and DSC (*n* = 9 vehicle-treated and *n* = 7 MCC950-treated). Mice used for CCl4 intoxification: *n* = 12 for N (*n* = 6 vehicle-treated and *n* = 6 MCC950-treated), *n* = 10 for %BW, B and DSC. (*n* = 3 vehicle-treated and *n* = 7 MCC950-treated). Mice used for pancreatic cancer: *n* = 14 for %BW, B, N, DSC and FCM (*n* = 7 vehicle-treated and *n* = 7 CHC and Met treated). Mice used for chronic pancreatitis: *n* = 16 for %BW, B, N, DSC and FCM (*n* = 8 vehicle-treated and *n* = 8 miRNA21 inhibitor treated).

## Data Availability

The data presented in this study are available at https://doi.org/10.6084/m9.figshare.21388989.

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
