# Peer review of "Generalizability, Robustness and Replicability When Evaluating Wellbeing of Laboratory Mice with Various Methods"

_animals, 2022, doi:10.3390/ani12212927_

Round 1

Reviewer 1 Report

The manuscript is well written. The authors has to include the information about the number of animals which data is used for the analyses for each intervention type. In the discussion I miss a part about the adaptability of the results on other species. Since I have the feeling that the results are specific to mice, I would include that in the title as well. In my opinion the lower well-being of animals (also humans) after a surgery, treatment or disease is not a wonder, and it is unavoidable. To me the question is can we set threshold values, to say that we should not cause higher reduction in wellbeing than that?

Author Response

We thank the reviewer for the very constructive suggestions. We marked all changes in the manuscript using track changes and answered comments in the attached point to point reply (please note that the line numbers refer to the manuscript when you fade out the deleted sentences).

Reviewer 2 Report

The manuscript entitled "Generalizability, robustness, and replicability when evaluating animal wellbeing with various methods" offers valuable information on how our results, when measuring animal welfare, are subject to sources of variation such as animal model type of procedure, disease, or distress. The findings reveal the need to refine our methods biologically and statistically but also invite us to acknowledge the heterogeneity and variability in animal experiments.

Assessing animal welfare reliably and robustly requires implementing complementary physiological and behavioral protocols coupled with multivariate statistical methods to increase our models' classification and prediction power. The receiver operating characteristic (ROC) has been considered the gold standard for assessing the performance of classifiers in biomedical and animal sciences. In this way, the manuscript goes ahead in a further step to refine animal welfare assessment.

The authors know the field and have earlier experience in animal welfare. They collected much information from previous experiments and carefully wrote the manuscript. The manuscript supplies information about how each method of animal welfare assessment can be more meaningful in specific circumstances (e.g., diseases, stimuli, procedures). 

My review is in the attached file. It holds several remarks, mostly minor comments or questions, pointing to the organization of the manuscript, the reason for selecting the ROC as a method for classification, the scope of the conclusions, and the organization of the references.

I encourage the authors to include the complete information from the discriminative power analyses in the figures or supplementary figures/tables. AUC values plus 95% confidence intervals are more meaningful than uniquely p-values, especially when, for example, we have a small number of animals. AUC, like every method, has intrinsic caveats and limitations, which you should address in the manuscript. Finally, I invite the authors to acknowledge that each model may be distinctly associated with distress/disease/responses; therefore, differences in results can be biologically valid.

Author Response

(The authors gave the same response as above.)

Reviewer 3 Report

The paper by Zechner et al describes a method and its application to estimate the replicability, robustness and generalizability of putative measures of mouse wellbeing after experimental manipulations. Analyses are performed using areas under the curve (AUCs). This approach is novel for welfare evaluation in animal experimental studies. The concept is introduced well, and the assessment of its usability is performed using data from already performed and published studies. This is an approach in the spirit of the 3Rs, as no additional animals were used.

Studies in which mice were subjected to surgery (e.g. implantation of a telemetric transmitter), or in which the animals served as model or gastrointestinal disease were re-analyzed.  The replicability, robustness and generalizability of the putative wellbeing indices differed between surgery conditions and disease models. The conclusions are based on the p-values associated with the calculated AUCs.

Considering that group sizes were different and in some instances very small (down to n=3), I would have favored a somewhat more cautious approach, e.g. by correcting the p-values for multiple comparisons of by applying a correction for the False Discovery Rate. The authors should consider this suggestion and the problem of small Ns, and should address these points  in more detail in the Discussion.

I also suggest to add a Box to the manuscript in which the concept of using AUCs and their calculation and interpretation are explained in a more fundamental and educational way, eventually supplemented with a simple set of (fictive) data, that is used for calculation of the AUC and the testing against the hypothesis that the measures have no diagnostic power (due to a  diagonal line of the AUC).

Specific comments and questions

Lines 62 ff.: I suggest to explain the ROC curves, using appropriate figures, in a separate box. Although readers of this paper should be familiar with ROCs, their use for the evaluation of wellbeing is new and deserves a good, fundamental  explanation (the present text already provides a good introduction, but the text might be made more "educational". Also, an example with a very simple set of data may be used to show how to calculate the ROC curve and how to determine specificity.

Line 110: When was the light switched ON and OFF?

Line 113: Replace "Deutschland" by "Germany"

Line 122: Report size and weight of the implanted device.

Line 148: In general: don’t only report the doses, but also report the injection volumes.

Lines 174-205: I suggest to try to depict the scheme of measurements in ana additional graph on a common time scale separately for the different interventions and disease models.

Lines 206 ff.: In this paper, different putative measures of wellbeing are compared, and the p-values are taken as index for specificity and discriminative power. I wonder whether a correction for multiple comparisons (or calculation of the false discovery rate) should be applied to these data. In particular, some of the p-values are based on very low group sizes (down to n=3).

Line 213: : (…), differences with p < 0.01 were considered to be highly significant.” Meaning what? The p-value is no reflection of the size of an effect. This is why I would like to see the values for AUC and their Cis in the different tables, and not only the p-values.

Line 228: It is easy to add the number of days to "pre" and "post" on the x-axis. of panels A-C of Figure 1

Lines 248-250: Are these behavioral measures relevant? The implanted device may have hindered the animals to perform, e.g. burrowing and nesting, simply because of the surgery and to their size in the animals abdomen.

Figure 2: Instead of only reporting the p-value, I also would show the AUC and Ci (perhaps printed smaller under the p-value).

Lines 285-288:: “(…) healthy and distressed animals was dependent on the specific surgical intervention. Alternatively, it might also be possible that the predictive power of the methods cannot be replicated, even when the same surgical intervention is performed.” I wonder whether it was dependent on physical restrictions as a direct consequence of the surgery.

What might be replicable is that some of the measures believed to reflect wellbeing don't have predictive power.

Lines 294-295: Please rephrase this sentence.

Figure 3: Instead of only reporting the p-value, I also would show the AUC and CI (perhaps printed smaller under the p-value).

Line 351: “in specific animal models for gastrointestinal animal models.”  Please rephrase.

Figure 4: Instead of only reporting the p-value, I also would show the AUC and CI (perhaps printed smaller under the p-value).

Lines 364-366: Cannot a deviation of body weight compared to pre-treatment values reflect abnormal processes that indicate effects on wellbeing?

Lines 366-372: The sizes of some treatment groups are very low. Be careful with conclusions (see also my comments to line 206 ff.).

Figure 5: Instead of only reporting the p-value, I also would show the AUC and CI (perhaps printed smaller under the p-value).

Lines 400-401: Weren’t the mice tested at about 4 months of age? How likely is it that mice of this age show an increase in body weight over the testing period?

Line 405 ff. (Discussion): Please also address the problem of multiple comparisons in relation to the p-values obtained, and of low group sizes in the discussion.

Physical constraints by implanted devices may also affect (some of) the measures used to assess wellbeing.

Which additional measures may be considered for evaluating the effects of experimental intervention in mice?

At which time points should effects on wellbeing be assessed (before and after intervention)?

Considering the discussion about the "replication" crisis of animal experimental data, what makes the authors so confident that all their results are reproducible and generalizable?

Author Response

(The authors gave the same response as above.)
